# DOPRA: Decoding Over-accumulation Penalization and Re-allocation in Specific Weighting Layer

Jinfeng Wei
Northeastern University
Language and Intelligent Systems Laboratory
Shenyang, P. R. China
202119033@stu.neu.edu.cn

Xiaofeng Zhang*
Shanghai Jiao Tong University
School of Electronic Information and Electrical
Engineering
Shanghai, P. R. China
framebreak@sjtu.edu.cn

## Abstract

In this work, we introduce DOPRA, a novel approach designed to mitigate hallucinations in multi-modal large language models (MLLMs). Unlike existing solutions that typically involve costly supplementary training data or the integration of external knowledge sources, DOPRA innovatively addresses hallucinations by decoding specific weighted layer penalties and redistribution, offering an economical and effective solution without additional resources. DOPRA is grounded in unique insights into the intrinsic mechanisms controlling hallucinations within MLLMs, especially the models' tendency to over-rely on a subset of summary tokens in the self-attention matrix, neglecting critical image-related information. This phenomenon is particularly pronounced in certain strata. To counteract this over-reliance, DOPRA employs a strategy of weighted overlay penalties and redistribution in specific layers, such as the 12th layer, during the decoding process. Furthermore, DOPRA includes a retrospective allocation process that re-examines the sequence of generated tokens, allowing the algorithm to reallocate token selection to better align with the actual image content, thereby reducing the incidence of hallucinatory descriptions in auto-generated captions. Overall, DOPRA represents a significant step forward in improving the output quality of MLLMs by systematically reducing hallucinations through targeted adjustments during the decoding process.

## CCS Concepts

• **Computing methodologies → Natural language processing**; **Computer vision**.

## Keywords

Multimodal Large Language Model, Over-accumulation, Attention layer

---

* represents the corresponding author, Xiaofeng Zhang is the corresponding author, his email:framebreak@sjtu.edu.cn.

---

**ACM Reference Format:**
Jinfeng Wei and Xiaofeng Zhang*. 2024. DOPRA: Decoding Over-accumulation Penalization and Re-allocation in Specific Weighting Layer. In *Proceedings of the 32nd ACM International Conference on Multimedia (MM '24), October 28-November 1, 2024, Melbourne, VIC, Australia.* ACM, New York, NY, USA, 10 pages. https://doi.org/10.1145/3664647.3681076

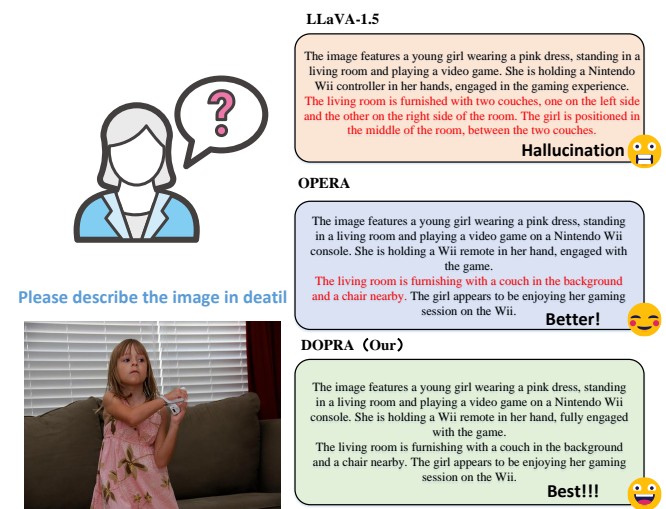

**Figure 1: Compare results of LLaVA-1.5 with DOPRA and OPERA.**

## 1 Introduction

Recently, Multimodal Large Language Models (MLLMs)[1, 2, 13, 15, 32, 38, 48, 55, 58, 63] have made groundbreaking advancements, fundamentally altering the way AI interacts with visual data and significantly enhancing fluent communication based on image semantic content. Despite their remarkable performance in handling a range of visually-centered tasks [4, 7, 8, 25, 29, 30, 57], understanding complex contexts[23, 59], or generating coherent narratives [4, 6, 25], MLLMs still grapple with a profound challenge: the "hallucination" problem. This refers to instances where MLLMs generate inaccurate or disjointed responses to visual inputs by incorrectly identifying nonexistent objects, attributes, or relationships within provided images. Such errors carry significant risks, particularly in high-stakes applications like autonomous driving [41, 45], where

misinterpreting visual cues could lead to life-threatening situations. While numerous methods [36, 42, 49, 52, 62] have been proposed to tackle hallucination issues, these often require costly interventions such as fine-tuning with annotated data [28], incorporating auxiliary models, or leveraging external knowledge sources.

This paper delves into addressing the hallucination conundrum during MLLMs' reasoning process without relying on supplementary data, external models, or specialized knowledge. Our investigation stems from a novel discovery related to what we term as "summary tokens" in the generation sequence, where attention weights accumulate early on. Analogous to recently discovered "anchor tokens" in the NLP domain [50], our analysis of self-attention graphs reveals a recurring pattern that frequently follows the generation of tokens with columnar attention structures, often leading to hallucinatory content. These summary tokens themselves tend not to carry substantial informational content (such as punctuation). However they appear to play a critical role in aggregating prior knowledge and guiding subsequent sequence generation.

The reliance on this aggregation pattern seems to induce hallucinations in contemporary MLLMs. To delve deeper into this phenomenon and provide a visual representation of the existence and impact of "summary tokens" in the generated sequences, we conducted theoretical analyses of their potential roles in text generation and detailed visualization of all tokens' self-attention weights. The experimental results show that there indeed exist specific tokens with disproportionately high attention weights relative to others. These high-weight tokens act as pivotal hubs, condensing the core meaning from preceding generated content. Typically, visual-related tokens are placed at the beginning of the sequence to ground the model's response in visual comprehension. However, as the generated text grows longer, visual information can become diluted through these summary tokens since they fail to adequately encapsulate the visual context's entire richness. Consequently, later tokens may overly depend on recent summary tokens while disregarding initial image-representative markers, thereby giving rise to model-bias-induced hallucinations; for example, inferring the presence of a "car" or "person" merely from the mention of "road" earlier in the sentence. Moreover, an increased number of summary tokens tends to exacerbate the likelihood of MLLM hallucinations.

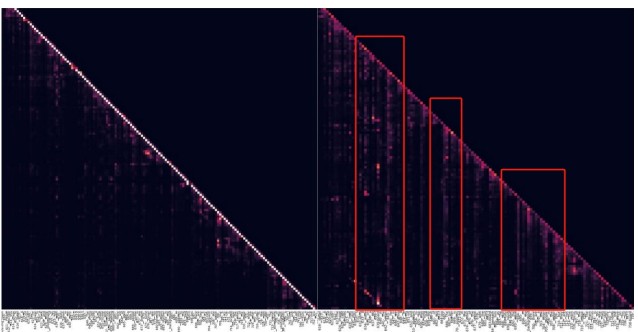

**Figure 2: Attention weighting graph comparison. Layer 8 weight maps are shown on the left and layer 12 weight maps are shown on the right.**

An example of LLaVA's hallucinations can be seen in Fig. 1. After experiments, we find that this kind of weight stacking actually exists in transformer's middle layer (for example, 12-20 layers), which is called "premature stacking".

Upon further scrutiny of the visualized attention weight maps, it is revealed that these highly accumulated weights do not develop gradually towards the end of decoding but instead start accumulating relatively early in the generation sequence. As shown in Fig. 2, the accumulation of attentional weights becomes very obvious from layer 12 onwards, accumulating much more than layer 8.

To tackle this issue of "premature stacking," we introduce DO-PRA, an innovative decoding framework built around two core strategies: Decoding Over-accumulation Penalization at specific attention layers and Re-allocation. DOPRA ingeniously integrates over-accumulation penalties into the beam search [5, 16, 47] process by applying weighted scores to candidate selections, effectively preventing tokens that exhibit strong patterns of over-trust. Specifically, for each decoding token, DOPRA inspects its local self-attention window, devising a column-wise metric to measure the strength of knowledge aggregation patterns and adjusts the model's log-probabilities accordingly to penalize the emergence of such patterns.

In addition, recognizing the persistence of knowledge aggregation patterns and the potential for hallucinations to permeate all candidate generations, DOPRA implements a retrospective reallocation strategy. This strategic retreat involves rolling back the decoding process to the position of summary tokens and judiciously reselecting candidates that bypass excessive accumulation patterns. Once an accumulated penalty score reaches a predefined threshold within the attention window, this rollback mechanism is triggered.

In this study, we specifically focus on vividly demonstrating the intrinsic connection between generated text tokens and their corresponding image attention regions. To this end, we visualize the top 50 most relevant highly responsive regions in the generated text (see Fig. 6). By combining textual information with visual embeddings, we are able to efficiently generate representative tokens for LLM that capture key visual features. Finally, by generating heat maps, we can visually check whether the generated text matches well with the corresponding image content. This approach not only gives us insight into how the model integrates textual and visual information during the generation process, but also allows us to clearly identify which visual elements play a decisive role in the generation process. Through this methodology, we not only deepen our understanding of the intrinsic working mechanism of multimodal language models, but also visualize the interactions between the generated text tokens and the attentional high-response regions of the images, thus enhancing the transparency of the model and the interpretability of the generated content. We conduct extensive empirical evaluations on benchmark datasets, employing hallucination-specific metrics, and testing advanced MLLMs, thereby substantiating DO-PRA's effectiveness in universally reducing hallucinations across various MLLM architectures. Our contributions can be summarized as follows:

- DOPRA presents an innovative solution that addresses hallucination issues in MLLMs during inference without requiring

external data, knowledge repositories, or additional training procedures.

- Through meticulous examination, DOPRA identifies the critical role played by summary tokens in the formation of hallucinations and develops a penalty-based decoding technique augmented with a backtracking reallocation strategy to disrupt excessive accumulation dynamics.
- Comprehensive evaluations demonstrate DOPRA's superior performance, proving it to be a practically cost-free intervention that effectively mitigates hallucinations in multimodal language models, thereby enhancing the credibility and reliability of these powerful AI tools in real-world applications.

## 2 Related Work

### 2.1 MLLM Development and Capacity

In recent years, Multimodal Large Language Models (MLLMs)[1, 2, 9, 10, 13, 32–34, 38, 48, 51, 63] have rapidly emerged and become the focus of both academia and industry. Since 2021, a series of representative models such as CLIP [43] and BLIP [26, 27] have pioneered large-scale pre-trained multimodal models, demonstrating the powerful ability to deeply integrate natural language with visual information, enabling accurate image description, cross-modal reasoning, and other functions. Entering 2023, this field is even more explosive, including but not limited to GPT-4V [1], LLaVA [38], minGPT-4 [63], InstructBLIP [13], Qwen-VL [2], CogVLM [51], and many other new multimodal macromodels have emerged one after another, which further enhance the model's ability of understanding and generating multimodal inputs, and make the MLLM get closer and closer to the general-purpose Artificial Intelligence [3, 18] in its ideal form.

### 2.2 Two-Track Path to MLLM Development

The current research and development of multimodal large models show two paths. On the one hand, researchers focus on continuously expanding the training dataset and model parameter sizes of the models in the hope of significantly improving their accuracy and generalization performance by increasing the model capacity. The new generation of multimodal large models, such as LLaVA-1.6-34B [38], GPT-4V [1], InstructBLIP [13], etc., are strong examples of this development trend, which have demonstrated their excellent capabilities in various complex multimodal tasks by virtue of their large number of parameters and rich training resources. However, although such large models continue to break the performance ceiling, the theoretical and technical challenges behind them cannot be ignored, especially in terms of resource consumption and computational efficiency. On the other hand, another research direction is to explore the intrinsic potential of small multimodal models [11, 54, 61, 64], and seek to achieve comparable or even similar functional effects as those of large-scale models in a smaller parameter space. The goal of this path is to optimize the model structure and training methods so that it can perform efficient multimodal understanding and generation under limited hardware conditions and computational costs. Although some breakthroughs have been made in such efforts, even the increasingly sophisticated small-scale models are still unable to completely escape from the "hallucination", a core problem shared by large multimodal models.

## 2.3 Strategies for Solving the MLLM Hallucination Problem

The term "hallucination" [17, 22, 31, 37, 40, 56] refers to the fact that multimodal models, when processing multimodal inputs, sometimes produce content that does not correspond to the actual inputs or is even fictitious. Aiming at this key problem, which seriously affects the credibility and practicality of models, the academic community has carried out a lot of fruitful research work and proposed several innovative solutions. Among them, RLHF [46, 53, 60] (Reinforcement Learning from Human Feedback) is an approach that relies on human feedback reinforcement learning techniques, which manually evaluates and guides model outputs, prompting the model to pay more attention to factual basis and logical consistency in the subsequent generation process. DoLa [12] improves the model's ability to capture and reproduce factual information when generating text by comparing the decoded information at different levels within the model. The Woodpecker [52] takes a two-stage approach to hallucination, first optimising the model initially with DINO [39] (a self-supervised learning framework), and then correcting the model's multimodal outputs by combining it with image captioning techniques. The OPERA [20] research team proposes a novel strategy to effectively mitigate the false inferences and hallucination representations generated by multimodal large models due to over-trusting one modality when processing complex scenes by applying the Over-Trust Penalty and Retrospection Allocation mechanisms. In the specific application scenarios of visual-linguistic modelling, the VCD [24] technique helps to identify and correct the object hallucination that may occur when the model is describing or reasoning about an image by introducing the visual contrast decoding link. In conclusion, whether following the traditional route of increasing model size or seeking the innovative path of efficient small-scale models, suppressing the "hallucination" phenomenon of large multimodal models has become an important issue of common concern to researchers, and substantial progress has been made in several cutting-edge researches. These strategies not only enrich the design and optimisation of multimodal models, but also lay a solid foundation for the construction of more accurate, reliable and universal multimodal intelligent systems in the future.

## 3 Method

In this section, we will first show the model generation process by introducing the modeling framework of MLLM. Then we will talk about DOPRA's Over-Accumulation Penalization and Re-allocation Decoding Strategies as shown in Fig. 3. Finally, it will be shown how high response works.

### 3.1 Anatomy of MLLM generation mechanism

**Integration of Input Constructs.** For the input construction phase of MLLM, the core is integrating two types of data sources: image and text. Regardless of the specific architecture, such models usually employ a visual coder to extract visual element features from the original image, and then transform and incorporate these features into the input space of the language model through a cross-modal mapping mechanism. We label this set of transformed visual elements as the set $x_v = \{x_0, x_1, ..., x_{N-1}\}$, where $N$ represents the number of visual elements and is fixed in many cases. Meanwhile,

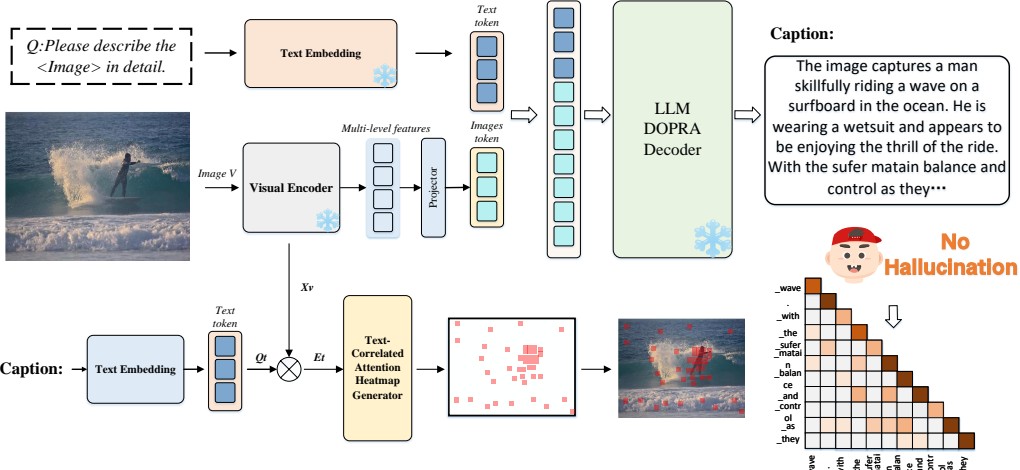

**Figure 3: The structure of Our method. The decoding method uses our proposed DOPRA. "Text-Correlated Attention Heatmap Generator" performs heatmap generation for $E_t$, the pseudo-code of which we put into the supplementary material.**

the textual input is processed by the disambiguation technique and is represented as the sequence $x_p = x_N, x_{N+1}, ..., x_{M+N-1}$. Finally, these two types of tokens are sequentially spliced to form the complete input sequence $\{x_i\}_{t=0}^{T-1}$, where $T$ is the sum of the total number of tokens of the image and text, $T = N + M$.

**Model forward propagation.** The MLLM follows an autoregressive model for training and uses a causal attention mechanism. Within the model, each token predicts the immediately following token based on the information of all the tokens preceding it, which is mathematically formulated as:

$$h = MLLM(\mathbf{x}_i)$$
$$h = \{h_0, h_1, \ldots, h_{T-1}\} \quad (1)$$

Here, $h$ is the sequence of hidden state vectors output by the last layer of the MLLM. Next, the model uses the vocabulary header mapping function $H$ to map the hidden state into a probability distribution for the next token prediction:

$$p(x_t|x_{<t}) = SoftMax[H(h_t)]_{x_t}, \quad x_t \in X \quad (2)$$

Here $\mathbf{x}_{<t}$ is a compact representation of all previous tokens, and $X$ represents the entire vocabulary set.

**Diverse Decoding Strategies.** Based on the predicted probability distribution $p(\mathbf{x}_t|\mathbf{x}_{<t})$ of each token obtained from the above computation, various decoding algorithms have been developed in the industry, such as the Greedy Decoding Method, the Beam Search algorithm, and advanced decoding strategies such as DoLa [12] and OPERA [20]. In the actual generation process, new tokens decoded at each step are added to the end of the original input text as the starting point for the next round of generation. OPERA visualizes the last layer of self-attention weights of the generated content by visualizing the last layer of self-attention weights. It develops a penalty strategy to enable it to reduce the impact of excessive accumulation of attention weights. However, there is a shortcoming that OPERA only investigates the occurrence of stacking at layer 32. We found that the accumulation of attention weights is not only

in 32 layers, but also in 32 layers, and the hallucination is actually generated "early".

## 3.2 "Over-accumulation" Attention-based Penalty

As shown in Fig. 4, in addressing the issue of delayed manifestation—whereby patterns indicative of over-reliance on potentially hallucinated information emerge only after several tokens have been decoded—we introduce the "Over-accumulation Attention Penalty" mechanism. This approach cumulatively applies a penalty to the beam search scores during generation, this method selectively targets and penalizes the accumulation of attention weights in a specified layer, particularly the 12th layer, influencing both the current token selection and the overall candidate sequences, thereby reducing the likelihood of selecting outputs containing hallucinations.

To realize this concept, we focus on the self-attention weights in the 12th layer local context window. Consider the generated sequence until time step $t$, denoted as $\{x_i\}_{i=0}^{t-1}$, and the causal self-attention weights used in Layer 12 to predict the next marker $\{\omega_{t-1,j}\}_{j=0}^{t-1}$ pertaining to the next token prediction, where $\omega = SoftMax\left(\frac{QK^\top}{\sqrt{D}}\right)$, and $Q, K, D$ represent query, key features, and the feature dimension, respectively. To capture the accumulation of knowledge aggregation patterns, we define a layer-specific local window of attention $W_k^{t-1}$ for layer 12 as follows:

$$W_k^{t-1,a=12} = \{w_i^a\}_{i=t-k}^{t-1}, \text{where } w_i^a = \{\omega_{i,j}^a\}_{j=t-k}^{t-1} \text{ and } a = 12 \quad (3)$$

Here, $k$ represents the size of the local window cropped from the attention map, and $\omega_{i,j}$ is the attention weight given by token $j$ to token $i$. Notably, we exclude attention weights from image tokens or prompt tokens, focusing exclusively on generated tokens ($t - k \geq N + M$). Additionally, we take the maximum attention

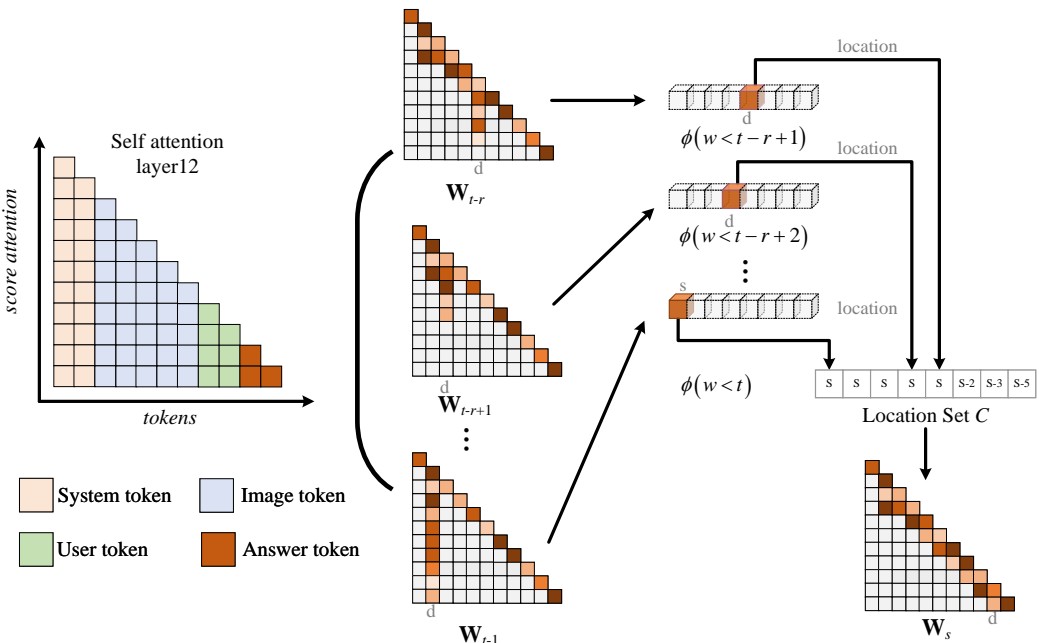

**Figure 4: The flow chart of DOPRA's decoder. The tokens of LLaVA1.5 are divided into system token, image token, user token and answer token. DOPRA carries out attention accumulation penalty for answer token.**

weight across multi-heads and renormalize these values as they often reflect high model confidence.

Upon obtaining $W_k^{t-1}$, we preprocess the attention weights by setting the upper triangle to zero and scaling up the remaining attention values for better representation, as shown in Equation :

$$W_k^{t-1,a=12} \triangleq \{w_i^a\}_{i=t-k}^{t-1}, \text{where } w_i^a = \{\sigma\omega_{i,j}^a\}_{j=t-k}^{i+1}, \text{for } a = 12 \tag{4}$$

The scaled attention matrix's lower triangle undergoes column-wise multiplication to generate a vector of scores, where higher scores suggest stronger knowledge aggregation patterns. The maximum value of this column-wise score vector serves as the pattern's descriptor:

$$\phi_{12}(\omega_{\leq t}^{12}) = \prod_{c=t}^{t-1} \sigma(w_{i,c}^{12}), \quad c = \arg\max_{t-k \leq j \leq t-1} \prod_{t-j}^{t-1} \sigma(w_{i,j}^{12}) \tag{5}$$

To efficiently apply the penalty without distorting the model's predictions towards unreasonable outputs, we form a candidate set $Y$ comprising the top-$N_{can}$ logits from each beam, where $|Y| = N_{can} \times N_{beam}$ and $N_{beam}$ is the number of beams. Then, we integrate the pattern metric $\phi(w_{\leq t})$ into the model logits to predict the next token while constraining it within $Y$:

$$p(x_t|x_{<t}) = Softmax[H(h_t) - \alpha_{12} \cdot \phi_{12}(w_{\leq t}^{12})]_x^t, \text{s.t. } x_t \in Y \tag{6}$$

Here, $w_{\leq t}$ summarizes all attention weights up to time step $t$, and $\alpha$ is a tunable hyperparameter controlling the strength of the

penalty applied to the logit. By introducing the "Over-accumulation Attention Penalty" with attention weight $\alpha = 12$ specifically emphasized, we aim to mitigate the risk of over-trust in potential hallucinations and guide the model toward more reliable generations.

## 3.3 Penalization and Re-allocation Strategy

Utilizing the "Over-accumulation" attention-based Penalty, we can proficiently detect the emergence of knowledge aggregation patterns after the generation of several consequent tokens. Ordinarily, the penalty discourages candidates exhibiting these patterns, thus encouraging the selection of others. Despite this, there are instances where all candidates are penalized even though hallucination has already taken place. This situation drives us to revisit the root cause of these aggregation patterns: they arise from early tokens excessively trusting the summary token, and the penalty fails to rectify this over-reliance. Therefore, a natural yet assertive solution is to eliminate tokens leading to hallucination and reallocate by choosing suitable tokens after the summary token. This leads us to propose the Retrospective Allocation strategy.

When the decoding process encounters a knowledge aggregation pattern and hallucination appears unavoidable, it reverses to the summary token and selects alternative candidates for the next token prediction, excluding those chosen earlier. The empirical condition for triggering retrospective decoding is based on the location overlap of the maximum values in the column-wise scores corresponding to multiple successive tokens, with a manually set threshold count denoted as $r$. Location counting proves more robust

and general compared to directly using the maximum value that varies across different models.

The complete retrospective process is detailed in Fig. 4. Leveraging the insights from Section 3.2, we can easily determine the location coordinate $c$ of the maximum score through Eq. (5). Following this, we establish the set of location coordinates for the recently decoded tokens $x_{t-l}, \ldots, x_{t-1}$, which is given by:

$$C = \{c : c = \arg \max_{t-k \le j \le z} \prod_{i=j}^{z} \sigma(w_{i,j}), z \in [t-l, t-1]\} \quad (7)$$

Here, $l > r$ is usually specified, and by default, we set $l = k$.

Given a sequence $\{x_0, x_1, \ldots, x_{t-1}\}$ and its associated recent location coordinate set $C$, we can assess the consistency of these coordinates. Formally, the overlap count is calculated as:

$$N_{overlap} = \sum_{c \in C} I_{c=s}, \text{ where } s = \text{Mode}(C) \quad (8)$$

$I$ is the indicator function returning 1 if the condition holds and 0 otherwise, and Mode gives the most frequent value in a set. If $N_{overlap} \ge r$, we initiate the retrospective action, considering $s = \text{Mode}(C)$ as the position of the summary token.

In the event that the sequence $\{x_0, x_1, \ldots, x_s, \ldots, x_{t-1}\}$ displays a knowledge aggregation pattern at the summary token $x_s$, the decoder rewinds to the subsequence $\{x_0, x_1, \ldots, x_s\}$ and selects a new next token from the complementary set $Y \setminus \{x_{s+1}\}$. To ensure progressive rollback, we require that the rollback location $s$ monotonically increases. Moreover, we impose a maximum rollback limit $\beta$; if $x_s$ reaches its rollback cap, we consider rolling back to $\{x_0, x_1, \ldots, x_{s-1}\}$.

## 3.4 High-Response Region Visualization and Cross-modal Interaction

To explore the relationship between generated text tokens and their correspondence with image attention in a more vivid manner, we visualize the high-response regions of the top 50 scores in Fig. 5. Given user input, we generate a text-guided query vector $Q_t \in \mathbb{R}^{M \times C}$, where $M$ denotes the number of queries. As shown in Fig. 3, this cross-modal interaction primarily occurs within the text decoder and can be readily instantiated using BERT [14] or Q-former [26] models. The generated text query $Q_t$ encapsulates salient visual cues that are most relevant to the user's command.

Employing the text query $Q_t$ along with the visual embeddings $X_v$, we can effectively generate representative tokens for the LLM that capture essential visual features. Specifically, the mixed attention mechanism aims to aggregate and condense visual features related to the text into a single context token. This process, depicted in Fig. 3, takes $Q_t$ and $X_v$ as inputs and formulates the mixed embedding of text and image, $E_t \in \mathbb{R}^{1 \times C}$, as:

$$E_t = Q_t \times X_v^T, \quad (9)$$

Contrary to Q-former, which employs 32 visual queries as LLM markers, our approach uses only the text query $Q_t$ to aggregate visual features with high response scores relative to the input command. Consequently, the compressed embedding $E_t$ efficiently retains the most critical visual cues associated with the user's input.

Finally, by normalizing $E_t$ and creating a heatmap, we can visually inspect whether the generated text corresponds well with the corresponding image content. Through this method, not only do we gain insight into how the model combines textual and visual information during generation, but we also clearly discern which visual elements play a decisive role in the generative process.

## 4 Experiment

### 4.1 Experimental Setup

**Decoding strategies.** Regarding decoding strategies, we have executed a variety of approaches for comparison and optimization. These include greedy decoding, which selects words at each step based on their highest probability; beam search [5, 16, 47] decoding with varying numbers of beams ($N_{beam}$ set as 5, 4, 3, 2, 1), allowing us to explore the impact of different search space widths; top-p nucleus sampling [19], using a standard setting of $p = 0.9$ to concentrate on the main body of the probability distribution; and the introduction of VCD [24] method that addresses object hallucination issues in large-scale models. For specialized decoding algorithms, like the DoLa [12] method designed to mitigate hallucinations in LLMs, within our experiments, we selected multiple candidate pre-mature layer indices ("0, 2, 4, 6, 8, 10, 12, 14") along with a fixed mature layer index at 32 to achieve fine-grained control over the internal decision-making process of the model. During the beam search decoding phase in the DOPRA experiment, we also set the scaling factor $\sigma$ to 50 to ensure effective discrimination of attention weights in the knowledge aggregation mode, where salient regions receive values greater than 1 while secondary areas get less than 1. Moreover, we established a default candidate number Ncan of 5, understanding that this hyperparameter can be adjusted but, in this experimental stage, we primarily study performance under default settings, considering that larger Ncan values could significantly increase the computational cost of the decoding process. **Implementation details.** Lastly, for other key hyperparameters influencing the decoding behavior in the LLaVA-1.5 model, we uniformly set $\alpha = 1$, $\beta = 5$, and $r = 15$, these configurations help maintain a stable experimental environment and allow us to focus on the effectiveness analysis of the proposed decoding strategies.

In summary, the DOPRA experiment delves deeply into exploring the performance boundaries of the LLaVA-1.5 model on multimodal tasks by employing a series of meticulously designed decoding strategies and parameter tuning. The aim is to validate the effects of different decoding techniques on the model's accuracy and robustness.

### 4.2 Quantitative Results

**DOPRA evaluation on hallucinations using CHAIR.** The Caption Hallucination Assessment with Image Relevance (CHAIR) [44] is a tailored metric for measuring object hallucination issues in image captioning tasks. For DOPRA, we utilize CHAIR to quantify the extent of hallucinated objects by computing the ratio of objects mentioned in the generated description that are absent in the ground-truth label set. CHAIR offers two separate assessments: $\text{CHAIR}_S$ (denoted as $C_S$) measures sentence-level hallucinations and $\text{CHAIR}_I$ (denoted as $C_I$) measures image-level hallucinations,

**Table 1: Compare results of DOPRA decoder with other decoder methods (baseline:LLaVA-1.5-7B).**

| Method | POPE ↑ | CHAIR$_S$ ↓ | CHAIR$_I$ ↓ |
|---|---|---|---|
| Greedy | **85.7** | 47.0 | 13.8 |
| Nucleus | 82.5 | 48.8 | 14.2 |
| Beam Search | 84.9 | 48.8 | 13.9 |
| OPERA [21] | 85.2 | 44.6 | 12.8 |
| DoLa [12] | 83.2 | 47.8 | 13.8 |
| VCD [24] | 84.5 | - | - |
| DOPRA | 85.6 | **42.4** | **12.3** |

mathematically expressed as:

$$C_S = \frac{|\text{hallucinated objects}|}{|\text{all mentioned objects}|}$$

$$C_I = \frac{|\text{captions w/ hallucinated objects}|}{|\text{all captions}|}$$

On the MSCOCO dataset [35], which encompasses over 300,000 images annotated with 80 objects, we perform CHAIR evaluations. Specifically, we randomly sample 500 images from the validation set of COCO 2014 and prompt different MLLM models with "Please describe this image in detail." to generate their descriptions. To ensure fairness, we restrict the maximum number of new tokens for generating descriptions of both long and short lengths. As displayed in Table 1, DOPRA demonstrates a clear superiority over baseline decoding methods in both $C_S$ and $C_I$ metrics. This advantage is consistent across both long and short description generations.

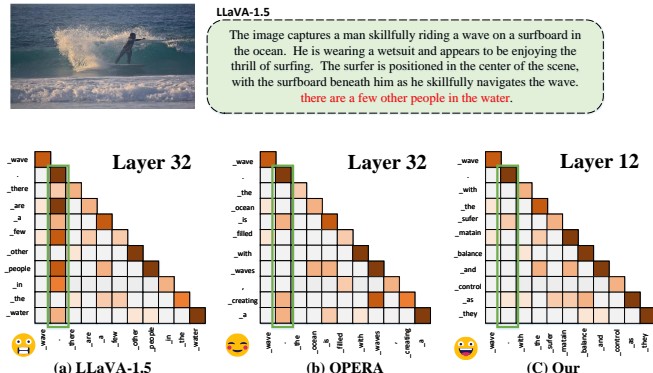

**Figure 5: Attention compare results of reason tokens.**

**DOPRA evaluation on hallucinations using POPE.** The Polling-based Object Probing Evaluation (POPE) [31] is a recently developed method aimed at assessing hallucination issues in MLLMs. Analogous to CHAIR, POPE scrutinizes object hallucination by prompting the model with an inquiry akin to "Is There a <object> in the image?" to gauge whether the model correctly identifies the presence or absence of a specific object in the given image. POPE includes three distinct evaluation scenarios: "random", "popular", and "adversarial".

**Table 2: Compare results of DOPRA with OPERA decoder on POPE and CHAIR dataset(N represents beam search number).**

| Method | Size | N | POPE ↑ | CHAIR$_S$ ↓ | CHAIR$_I$ ↓ |
|---|---|---|---|---|---|
| LLaVA1.5+OPERA | 7B | 1 | 85.4 | 48.6 | 14.7 |
| LLaVA1.5+OPERA | 7B | 2 | **85.5** | 48.4 | 14.5 |
| LLaVA1.5+OPERA | 7B | 3 | 85.4 | 48.4 | 14.1 |
| LLaVA1.5+OPERA | 7B | 4 | 85.3 | 49.5 | 14.5 |
| LLaVA1.5+OPERA | 7B | 5 | 85.2 | **44.6** | **12.8** |
| LLaVA1.5+DOPRA | 7B | 1 | 85.7 | 48.4 | 14.5 |
| LLaVA1.5+DOPRA | 7B | 2 | **85.8** | 48.4 | 14.4 |
| LLaVA1.5+DOPRA | 7B | 3 | 85.5 | 48.4 | 14.1 |
| LLaVA1.5+DOPRA | 7B | 4 | 85.4 | 49.4 | 14.4 |
| LLaVA1.5+DOPRA | 7B | 5 | 85.6 | **42.4** | **12.3** |
| InstrcuBlip+OPERA | 7B | 1 | **84.7** | 48.5 | 15.5 |
| InstrcutBlip+OPERA | 7B | 2 | 84.3 | 48.4 | 15.3 |
| InstrcutBlip+OPERA | 7B | 3 | 84.2 | 48.5 | 15.1 |
| InstrcutBlip+OPERA | 7B | 4 | 84.2 | 49.2 | 15.2 |
| InstrcutBlip+OPERA | 7B | 5 | 84.6 | **46.7** | **14.4** |
| InstrcutBlip+DOPRA | 7B | 1 | **85.3** | 48.2 | 15.1 |
| InstrcutBlip+DOPRA | 7B | 2 | 84.7 | 48.1 | 14.9 |
| InstrcutBlip+DOPRA | 7B | 3 | 84.7 | 48.0 | 14.8 |
| InstrcutBlip+DOPRA | 7B | 4 | 84.7 | 48.7 | 14.8 |
| InstrcutBlip+DOPRA | 7B | 5 | 85.1 | **46.1** | **14.0** |
| MiniGPT4+OPERA | 7B | 1 | **73.3** | 28.7 | 10.2 |
| MiniGPT4+OPERA | 7B | 2 | 72.4 | 26.9 | 10.0 |
| MiniGPT4+OPERA | 7B | 3 | 71.6 | 26.8 | 9.7 |
| MiniGPT4+OPERA | 7B | 4 | 71.8 | 26.8 | 9.9 |
| MiniGPT4+OPERA | 7B | 5 | 72.8 | **26.2** | **9.5** |
| MiniGPT4+DOPRA | 7B | 1 | **75.6** | 19.7 | 9.1 |
| MiniGPT4+DOPRA | 7B | 2 | 73.6 | 19.3 | 9.0 |
| MiniGPT4+DOPRA | 7B | 3 | 72.0 | **19.2** | 8.9 |
| MiniGPT4+DOPRA | 7B | 4 | 72.1 | 24.9 | 8.9 |
| MiniGPT4+DOPRA | 7B | 5 | 73.2 | 25.8 | **8.8** |

We validate DOPRA using POPE on LLaVA [38] MLLM models and report the mean F1 scores in Table 1 and Table 2. When compared with baseline decoding strategies, DOPRA also achieves the best performance, although the improvements might be marginal. It is important to note that DOPRA particularly excels in mitigating hallucinations in longer sequences. In the context of POPE answers, which tend to be brief responses starting with "Yes" or "No" followed by confirmations like "Yes, there is a <object> in the image.", the knowledge aggregation patterns—central to our method—may not surface as conspicuously. Nevertheless, DOPRA still demonstrates a competitive edge in controlling hallucinations across various lengths of sequences and different evaluation contexts.

In this section, we assess DOPRA's effectiveness in reducing hallucinations in both extended descriptions and concise VQA answers as shown in Table 1. We show the difference between DOPRA and OPERA in attention punishment in Fig. 5. It can be seen that OPERA only carries out attention punishment in the last layer of Transformer, namely layer 32, while we observe early attention accumulation in the middle layer of transformer between layer 12 and 18, so we choose to intervene in layer 12. Our intervention

measures are more consistent with the "early phenomena" of the model than OPERA, and our indicators are better on the POPE and CHAIR data sets.

## 4.3 High Response Example Analysis

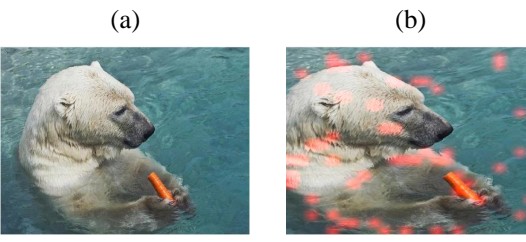

**Figure 6: High response visual results of tokens (a is the input image, b is the high repsonse results by caption).**

As shown in Fig. 6, in this section, we delve into the significant advancements and core mechanisms within the realm of caption generation models—specifically focusing on the DOPRA algorithm—in addressing and rectifying caption hallucination phenomena. We begin by presenting an extensive examination of real-life cases where caption hallucinations have been effectively corrected using DOPRA, juxtaposing pre- and post-correction captions in situations such as "is there any dog?", thereby illustrating DOPRA's substantial improvements in accurately capturing and conveying the actual content of images.

Subsequently, we explore from the perspective of attention mechanisms how DOPRA systematically refines its strategy for allocating attention weights to precisely latch onto key visual features during caption generation. This includes, among other things, displaying different levels of attention maps and their evolution throughout model optimization, highlighting how the model learns to disregard irrelevant information and hone in on decisive visual cues within an image, say, when discerning whether or not a dog is present.

To further unravel the inner workings of DOPRA's efficacy in enhancing caption veracity, we conduct a detailed investigation into the genesis and case studies of high-response regions. Using visualization techniques, we demonstrate visually how the model responds differently to various areas of the input image, particularly pinpointing those critical regions with exceedingly high response values. These high-response zones not only reveal the pathways the model follows in recognizing and interpreting image content but are also vital for understanding the information extraction and decision-making processes involved in generating accurate captions.

## 5 Discussion and Limitations

The illusions generated by Large Language Models (LLMs) may result from inadequate generalization or the model's knowledge not being updated in a timely manner. We believe the illusion issues in Multimodal Large Language Models (MLLMs) are primarily attributed to the visual component, possibly due to the coarse granularity of features (a problem that also needs addressing, as current architectures like Q-Former tend to lose much information), or possibly due to the use of noisy data during the alignment phase, or insufficiently detailed alignment.

The illusion observed in captions is merely a superficial phenomenon; the root cause lies in the process of encoding by the vision encoder, which is essentially a compression process, leading to a significant loss of original image information. This fundamental flaw in the architecture prevents effective captioning based on visual hidden representations and might even lead to illusions. This is because the vision encoder, CLIP, is trained on images and highly noisy captions.

Our overarching idea is that DOPRA is merely a temporary solution, and there will undoubtedly be a need for end-to-end finetuning and improvements in perceptual capabilities in the future. Future research may explore avenues such as implementing more meticulous data alignment procedures that go beyond aligning entire images and texts at a superficial level, or incorporating fine-grained visual features to overcome the perceptual limitations of CLIP, which predominantly focuses on high-level semantic understanding rather than nuanced visual details.

## 6 Conlusion

In conclusion, DOPRA introduces a novel and cost-effective approach to address the prevalent issue of hallucination in multimodal large language models (MLLMs), thereby enhancing their precision and reliability for practical applications. By innovating a method that penalizes decoding over-accumulation and reallocates attention in specific weighting layers, DOPRA circumvents the need for additional training data or the integration of external knowledge bases, setting it apart from existing methods. This technique is grounded in a deep understanding of the mechanisms underlying hallucinations in MLLMs, particularly the disproportionate reliance on summary tokens to the detriment of image-relevant information in certain critical layers. Through the application of a weight stacking penalty and a strategic reallocation during the decoding phase, DOPRA effectively broadens the context considered by the model, while ensuring that generated captions more accurately reflect the authentic content of images. The implementation of this method marks a considerable advancement in the development of MLLMs by systematically reducing the occurrence of hallucinations, thereby improving the overall output quality of these models. DOPRA's innovative approach not only signifies a significant leap towards resolving a stubborn challenge in the field of artificial intelligence but also paves the way for future research and development in enhancing the interpretative and generative capabilities of multimodal systems.

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
