# OpenReview forum: "DOPRA: Decoding Over-accumulation Penalization and Re-allocation in Specific Weighting Layer"
_acmmm.org/ACMMM/2024/Conference — MM2024 Poster_

### Official Review · Reviewer_tUQR · 2024-05-24

**Rating:** 4
**Confidence:** 3

**Summary:**

This work introduces DOPRA, a interesting method to reduce hallucinations in multi-modal large language models (MLLMs). Unlike current approaches that require additional training data or external knowledge sources, DOPRA uses decoding-specific weighted layer penalties and redistribution. It addresses the issue of models overly relying on summary tokens and neglecting image-related information, particularly in certain layers like the 12th. By applying targeted penalties and a retrospective token allocation process, DOPRA effectively aligns token selection with actual image content, reducing hallucinations and improving output quality.

**Strengths:**

DOPRA presents a significant advancement in mitigating hallucinations in multi-modal large language models (MLLMs) through its cost-effective and efficient approach that avoids the need for additional training data or external knowledge sources. It uniquely employs decoding-specific weighted layer penalties and redistribution, addressing the intrinsic issues of models overly relying on summary tokens and neglecting crucial image-related information. By targeting specific layers, such as the 12th, DOPRA enhances precision during the decoding phase. Furthermore, its retrospective allocation process re-evaluates and reallocates token sequences to ensure better alignment with actual image content, effectively reducing hallucinations and improving the overall output quality of MLLMs.

**Limitations:**

1. Due to the overuse of the term "hallucinations," the concept has been extended to encompass various issues in MLLMs, such as responses that are inconsistent with user input, misalignment between responses and the contextual image-text information, and responses that contradict objective facts. If the innovations presented in this work can be effectively explained as addressing these types of hallucinations, it would provide a clear and compelling introduction to the study's contributions and innovations.
2. This work lacks a comparison with other parameter-adaptive hallucination mitigation solutions, particularly the ITI method [1], whose parameter attention activation design closely corresponds with this study. Incorporating a methodological and performance comparison with parameter-adaptive methods like ITI is necessary to validate the innovation and contributions of this research.

[1] Li, Kenneth, et al. "Inference-time intervention: Eliciting truthful answers from a language model." Advances in Neural Information Processing Systems 36 (2024).

**Suitability:**

3

---

### Official Review · Reviewer_UMJy · 2024-05-24

**Rating:** 4
**Confidence:** 3

**Summary:**

The paper presents DOPRA, a novel method designed to reduce hallucinations in Multimodal Large Language Models (MLLMs). It applies weighted overlay penalties in specific layers and includes a retrospective allocation process to re-examine and adjust token generation, thereby enhancing the accuracy and relevance of generated content. Experimental results demonstrate DOPRA's effectiveness in improving the output quality of MLLMs across various tasks without requiring additional data or external knowledge sources.

**Strengths:**

This paper proposed a novel approach to mitigate over-accumulation in self-attention layers, offering a fresh solution to hallucinations in Multimodal Large Language Models (MLLMs). The method's theoretical robustness is demonstrated through the integration of penalization and dynamic reallocation strategies, ensuring balanced attention distribution. Comprehensive evaluations validate its effectiveness, showing significant improvements without needing additional data. The paper is well-structured and clearly explains the methodology, making it accessible and comprehensible. DOPRA's broad applicability enhances the reliability and trustworthiness of MLLMs in various domains, such as natural language processing and image captioning.

**Limitations:**

More experimental tests involving combinations with various decoder methods, as well as LLaVA1.5 and MiniGPT4, are needed to demonstrate the superior performance of the proposed method.

**Suitability:**

2

---

### Official Review · Reviewer_W32S · 2024-06-03

**Rating:** 3
**Confidence:** 3

**Summary:**

This paper introduces DOPRA, a method designed to reduce hallucinations in multimodal large language models (MLLMs). The approach penalizes over-accumulation of attention weights in specific layers, and reallocates attention during the decoding process. The method effectively mitigates hallucinations without requiring additional training data or external knowledge sources. Empirical evaluations demonstrate that DOPRA significantly improves the accuracy and reliability of MLLM outputs compared to existing methods.

**Strengths:**

# Method Simplicity and Efficiency:
The proposed method is straightforward and based on a thorough analysis of the mechanisms within MLLMs. It effectively reduces hallucinations without requiring additional data or training.

# Comprehensive System Design:
The authors present a relatively complete and systematic design that not only includes methodological aspects but also incorporates visualization. This enhances the method’s acceptability and transparency.

# State-of-the-Art Results:
The method achieves state-of-the-art results across multiple benchmarks, demonstrating its effectiveness and robustness in fair comparisons.

**Limitations:**

# Motivation for Focusing on the 12th Layer:
The justification for concentrating on the self-attention weights in the 12th layer remains unclear. While the authors mention in the introduction that they observe a "premature stacking" phenomenon (lines 157-164), it is still not convincing why this specific layer is chosen. What about the 13th or 15th layers? Unfortunately, there is no ablation study addressing this aspect, either in the main manuscript or the supplementary materials.

# Originality and Clarity of Findings:
The findings and analysis appear to be derived from the paper "The Label Words are Anchors." Thus, the originality of these findings and the self-attention analysis seems weak. For readers who are not familiar with the referenced paper, understanding the entire concept might be challenging. It is suggested that the authors include a preliminary section to introduce the related findings more comprehensively.

# Presentation and Clarity:
The paper is not presented clearly. As noted in the first point, the motivation behind the proposed strategies and operations is not very clear. Additionally, it seems that the caption of the proposed method in Figure 1 is incorrectly placed, as it appears to be the same as that of OPERA, making it difficult to understand the superiority of the proposed method. Furthermore, Figure 2 becomes blurry when zoomed in, which affects the clarity of the information presented.

**Suitability:**

3

---

### Meta-Review · Area_Chair_rGsp · 2024-07-03

**Recommendation:** Accept (Poster)
**Confidence:** 5

**Metareview:**

This paper introduces a new and interesting method to mitigate over-accumulation in self-attention layers, and offers a fresh solution to hallucinations in Multi-modal Large Language Models (MLLMs). The proposed method innovatively utilizes decoding-specific weighted layer penalties and redistribution, which thus addresses the intrinsic issues of the models which overly rely on summary tokens while neglecting important image-related information. All the reviewers recommend acceptance.